# Untargeted and Targeted LC-MS/MS Based Metabolomics Study on In Vitro Culture of *Phaeoacremonium* Species

**DOI:** 10.3390/jof8010055

**Published:** 2022-01-06

**Authors:** Pierluigi Reveglia, Maria Luisa Raimondo, Marco Masi, Alessio Cimmino, Genoveffa Nuzzo, Gaetano Corso, Angelo Fontana, Antonia Carlucci, Antonio Evidente

**Affiliations:** 1Department of Clinical and Experimental Medicine, University of Foggia, Viale Pinto 1, 71121 Foggia, Italy; pierluigi.reveglia@unifg.it (P.R.); gaetano.corso@unifg.it (G.C.); 2Department of Agricultural Sciences, Food, Natural Resources and Engineering, Via Napoli 25, 71122 Foggia, Italy; marialuisa.raimondo@unifg.it; 3Department of Chemical Sciences, University of Napoli Federico II, Complesso Universitario Monte Sant’Angelo, Via Cintia 4, 80126 Napoli, Italy; marco.masi@unina.it (M.M.); alessio.cimmino@unina.it (A.C.); 4Institute of Bio-Molecular Chemistry, Consiglio Nazionale delle Ricerche (ICB-CNR), Via Campi Flegrei 34, 80078 Pozzuoli, Italy; nuzzo.genoveffa@icb.cnr.it (G.N.); afontana@icb.cnr.it (A.F.); 5Laboratory of Bio-Organic Chemistry and Chemical Biology, Department of Biology, University of Napoli Federico II, Via Cupa Nuova Cinthia 21, 80126 Napoli, Italy

**Keywords:** *Phaeoacremonium* spp., Esca complex disease, scytalone, isosclerone, LC-MS/MS

## Abstract

Grapevine (*Vitis vinifera* L.) can be affected by many different biotic agents, including tracheomycotic fungi such as *Phaeomoniella chlamydospora* and *Phaeoacremonium minimum*, which are the main causal agent of Esca and Petri diseases. Both fungi produce phytotoxic naphthalenone polyketides, namely scytalone and isosclerone, that are related to symptom development. The main objective of this study was to investigate the secondary metabolites produced by three *Phaeoacremonium* species and to assess their phytotoxicity by in vitro bioassay. To this aim, untargeted and targeted LC-MS/MS-based metabolomics were performed. High resolution mass spectrometer UHPLC-Orbitrap was used for the untargeted profiling and dereplication of secondary metabolites. A sensitive multi reaction monitoring (MRM) method for the absolute quantification of scytalone and isosclerone was developed on a UPLC-QTrap. Different isolates of *P. italicum*, *P. alvesii* and *P. rubrigenum* were grown in vitro and the culture filtrates and organic extracts were assayed for phytotoxicity. The toxic effects varied within and among fungal isolates. Isosclerone and scytalone were dereplicated by matching retention times and HRMS and MS/MS data with pure standards. The amount of scytalone and isosclerone differed within and among fungal species. To our best knowledge, this is the first study that applies an approach of LC-MS/MS-based metabolomics to investigate differences in the metabolic composition of organic extracts of *Phaeoacremonium* species culture filtrates.

## 1. Introduction

*Vitis vinifera* is one of the most economically important crops worldwide, with approximately 71% of the world’s grape production being used for wine production. A variety of fungal diseases threatens viticultural regions all over the world, compromising the yield and quality of wine [1,2]. Out of them, grapevine trunk diseases (GTDs), caused by one or several xylem-inhabiting fungi, produce a progressive decline in vines, consisting of a loss in productivity and eventually death of the vines [3]. Over the past few decades, they have been extensively studied [4]. However, the relationship between pathogenic fungi involved in GTDs and abiotic agents, the expression of symptoms, and the lack of effective management strategies, requires further investigation [5,6]. The most common GTDs include Esca and Petri diseases, Botryosphaeria, Diaporthe and Eutypa diebacks, and black foot disease [7,8]. In general, black-foot, Petri, and Botryosphaeria dieback are the most common GTDs affecting young vineyards, whereas Esca (in the strict sense) and Eutypa dieback are the typical GTDs affecting old vineyards. In this second case, Petri disease (Esca complex) and Botryosphaeria dieback can also frequently occur [9].

*Phaeomoniella chlamydospora*, and several *Phaeoacremonium* species [10,11,12] are the main fungal agents involved in the Esca complex. The *Phaeoacremonium* genus was originally described 24 years ago, containing only six species [13]. To date, the genus *Phaeoacremonium* includes 63 species distributed worldwide [14,15,16,17,18]. To date, 37 *Phaeoacremonium* species, including the most recent *P. adelophialidum* from Algeria, *P. album*, *P. junior* and *P. pravum* from South Africa, *P. canadense* and *P. roseum* from Canada, *P. italicum* from Italy and *P. nordesticola* from Brazil, have been isolated from grapevines showing Esca and Petri disease symptoms [13,16,17,19,20,21,22,23,24].

Secondary metabolites produced by fungal species are, in general, not essential for their growth but are believed to be advantageous under certain conditions and in distinct habitats, such as host colonisation for pathogenic fungi. Understanding their role in virulence remains an important challenge for chemists, molecular biologists, plant pathologists and physiologists. Indeed, secondary metabolites produced by fungal pathogens may play an important role in virulence and they may be linked to the symptoms observed on infected plants [25]. Furthermore, another fundamental role played by these metabolites is in the reciprocal interaction between host plant and fungi [25,26,27]. Furthermore, some of the secondary metabolites can be classified as fungal phytotoxins and they are usually divided into host-selective toxins (HSTs) and non-host selective toxins (NHSTs) [27]. In the last decades, several studies have been conducted on the isolation and characterisation of phytotoxic compounds produced in vitro by pathogens involved in GTDs [28]. Many of these metabolites have been chemically characterised and tested for their toxicity on grapevines and non-host plants, and different reviews and perspectives on these topic can be found in the literature [28]. Some of these compounds are typical for a specific pathogen. Scytalone (**1**) and isosclerone (**2**), two phytotoxic naphthalenone pentaketides, are typically produced by *Phaeoacremonium minimum* and *Phaeomoniella chlamydospora* involved in the Esca complex [29,30].

Investigating the in vitro metabolic production, obtained under different cultural or extraction conditions (pH, saline concentration, solvent polarity), of pathogenic fungi is one of the rational approaches in the identification of specific biomarkers involved in distinct metabolic pathway or stress response. Metabolites exhibit different physicochemical properties and belong to different chemical classes. Thus, different experimental conditions are needed to extract the whole metabolome [31,32].

Metabolomics is the latest of the “-omics” sciences and its capability of accelerating the discovery of active and novel metabolites from plants, fungi, and bacteria has increased in the last decades [33,34,35]. Since metabolomics, and in general –omics sciences, are based on multidisciplinary approaches, their application can encourage the creation of professional networks among scientists, paving the way to more systemic study. In consideration of the alarming frequency of fungal infections in medicine and agriculture, metabolomics appears an optimal tool to face the study of the virulence of fungal pathogens and improve biocontrol and treatment of these infections [36,37].

In particular, enormous progress in LC–MS-based metabolomics has been made, providing researchers with a variety of chromatographic separation, ionization, and mass analysers [38,39]. Moreover, it has also been applied in the dereplication of already known substances in the early stages of the screening process of bioactive mixtures [40,41,42].

Based on this background, the objectives of this study were: (i) to investigate the secondary metabolites produced by three *Phaeoacremonium* species and assess their toxicity by in vitro bioassay; (ii) to evaluate the effect of the pH in the extraction process; (iii) to develop a targeted MS method for the quantitation of known compounds produced by the selected fungi; (iv) to select *Phaeoacremonium* isolates for additional investigation.

## 2. Materials and Methods

### 2.1. Phaeoacremonium Species and Culture Conditions

Nine representative isolates belonging to three different *Phaeoacremonium* species were selected. In details, four isolates of *P. italicum* (CBS 137763, *extype* culture; Pm50M; Pm59; Pm45) were retrieved from the collection of the Department of Science, Food Natural Resources and Engineering (DAFNE), University of Foggia (Foggia, Italy). Three isolates of *P. alvesii* (CBS 113590; CBS 729.97; CBS 408.78) and two isolates of *P. rubrigenum* (CBS 498.94 *extype* culture; CBS 112046) were retrieved from Westerdijk Fungal Biodiversity Institute (CBS, Utrecht, The Netherlands). Taxonomic and morphological identity of all *Phaeoacremonium* species were confirmed and discussed according to Laidani et al. (2021) [43].

All nine isolates were grown under the following conditions: 150 mL of sterile Czapek broth (3 g/L, NaNO_3_; 1 g/L, KH_2_PO_4_·3H_2_O; 0.5 g/L, KCl; 0.01 g/L, FeSO_4_·7H_2_O; 1 g/L, yeast extract; 1 g/L, malt extract; 30 g/L, sucrose) seeded with 2 mL of a mycelia suspension of pure culture of each fungus, previously grown on potato dextrose agar (PDA) for 7–10 days at 23 ± 2 °C in the dark. The liquid culture was then incubated at 25 °C for 4 weeks in darkness.

### 2.2. Extraction Procedure

At harvest, the mycelial mat was removed from each flask by filtration under vacuum with sterile filter paper. Extractions were carried out at two different pHs: (1) unmodified pH (UpH) of culture filtrates (pH ranged from 4.4 to 8.8 among the species tested); (2) pH 2, by acidification with 1 M formic acid (Sigma-Aldrich, Milan, Italy). For each fungal isolate, two samples of 40 mL were taken from the culture filtrates and their pH modified. Then each culture filtrate was extracted three times using ethyl acetate (3 × 40 mL) (Sigma-Aldrich, Milan, Italy). Organic extracts corresponding to the same pH value were then combined, dried over Na_2_SO_4_ (Sigma-Aldrich, Milan, Italy), filtered, and evaporated under reduced pressure. The organic extracts obtained from each fungal isolate and pH value, and their corresponding aqueous phases, were tested for phytotoxicity as explained in the following section.

### 2.3. Phytotoxic Bioassays

The culture filtrates and the organic extracts (UpH and pH 2) were assayed on non-host cotyledons of *Cucumis sativus* L. Before the inoculation, the cotyledons were disinfected with EtOH 70% for 45 sec, subsequently rinsed three times with sterile distilled water and finally dried in sterile paper. For the culture filtrates, two droplets (20 μL) were directly spotted on each cotyledon without dilution. Droplets (20 μL) of sterile water and of Czapek broth were used as negative controls. Meanwhile, the organic extracts were tested at the concentration of 3 and 1.5 mg/mL. The organic extracts were dissolved in MeOH and diluted with distilled H_2_O up to the assay concentrations (the final content of MeOH was 4%). Droplets of the test solutions (10 μL) were applied on the axial side of cotyledons that had been needle punctured 3–4 times. Droplets (10 μL) of Czapek broth and MeOH in distilled H_2_O (4%) were applied on cotyledons as negative controls. Both experimental bioassays were repeated twice. The cotyledons were then kept in a moist chamber to prevent the drying of the droplets, observed daily and scored for symptoms after five and eight days. The presence of symptoms obtained by organic extract bioassays was evaluated by observation using a scale of 0–5, where 0 = no symptoms observed; 1 = 1–20%; 2 = 21–40%; 3 = 41–60%; 4 = 61–80%; and 5 = 81–100% of cotyledon surface showing necrotic symptoms. The data recorded were used to determine the overall toxicity severities (*TS*) according to Equation (1): (1)$$TS=\frac{\sum \left(Number\;of\;observ\times values\;of\;scores\right)}{Total\;number\;of\;cases}$$

### 2.4. Reference Standards

The scytalone (**1**) and isosclerone (**2**) that were used as standards were isolated from in vitro cultures of *P. minimum* [29] and *Neofusicoccum parvum* [44], respectively. As internal standard (I.S.) for the LC-MS/MS analysis, 2-hydroxy-1,4-naphthoquinone (**3**), purchased from Sigma-Aldrich, Milan, Italy, was used.

### 2.5. Untargeted and Targeted LC-MS/MS Analysis

Chromatographic separations were achieved by two LC–MS platforms based on either a Q-Exactive Hybrid Quadrupole-Orbitrap (Thermo Fisher, Waltham, MA, USA) coupled with an Infinity 1290 UHPLC System (Agilent Technologies, Santa Clara, CA, USA), or a hybrid triple quadrupole/linear ion trap tandem mass spectrometer (QTRAP 4500, AB Sciex, Framingham, MA, USA) coupled to an Eksigent Ekspert ultraLC UPLC. The same chromatographic conditions were applied to both platforms by using a C18 column (Eclipes Plus, 3.5 um, 4.6 × 100 mm) from Agilent (Santa Clara, CA, USA) at 28 °C and an elution program consisting of a linear gradient from 10% to 80% of 0.1% (*v*/*v*) formic acid in ACN containing 0.1% (*v*/*v*) formic acid in 6 min. A post run equilibration step of 4 min was included prior to each analysis. The injection duty cycle was 10 min considering the column equilibration time. Flow rate was 0.4 mL/min. The injection volume was 5 μL and the autosampler was maintained at 10 °C.

For the untargeted analysis, the Q-Exactive-based instrument was equipped with a Heated Electrospray (HESI) source with the following setting: spray voltage positive polarity 3.2 kV, negative polarity 3.0 kV, capillary temperature 320 °C, S-lens RF level 55, auxiliary gas temperature 350 °C, sheath gas flow rate 60, and auxiliary gas flow rate 35. Full MS scans were acquired over the mass range 150–500 with a mass resolution of 70,000. The target value (AGC) was 1 × 10^6^ and the maximum allowed accumulation time (IT) was 100 ms. For the data dependent MS/MS (ddMS2) analyses a Top10 method was used. The ten most intense peaks were selected for fragmentation with a stepped normalized energy of 25–28 and 20–30 in positive and negative ionization mode, respectively. AGC was 2 × 10^5^ with IT 75 ms and 17,500 mass resolution. Injection volume was of 5 μL.

Targeted MS analysis was performed on hybrid triple quadrupole/linear ion trap tandem mass spectrometer QTRAP 4500. Q1 resolution was adjusted to 0.7 ± 0.1 amu for multi reaction monitoring (MRM), referred to as unit resolution. Q3 was also set to unit resolution in MRM mode. MS analysis was carried out in positive ionization mode using an ion spray voltage of 4500 V. The nebulizer and the curtain gas flows were set at 30 psi using nitrogen. The Turbo V ion source was operated at 400 °C with the auxiliary gas flow (nitrogen) set at 50 psi. Two suitable multi reaction monitoring (MRM) transitions were selected for the compounds scytalone (**1**) and isosclerone (**2**), while one transition was selected for the I.S. 2-hydorxy-1,4-naphtoquinone (**3**). The compound dependent parameters for the three compounds were optimized using the manual optimization protocol in tuning mode. The Q1 mass, the Q3 transition and the best parameters are reported in Table 1.

Validation study was obtained analysing calibration curves, limit of detection (LOD), limit of quantification (LOQ), within-day and between-day imprecision and inaccuracy. Calibration curves were obtained reporting the Area ratio (Compound Area/I.S. Area) against the compound concentration. I.S. was spiked in every standard solution in a concentration of 5 μg/mL. Two different curves were built for scytalone (**1**) and isosclerone (**2**), respectively (Appendix A). Each solution was injected in triplicate. LOD and LOQ were calculated as a magnitude of, respectively, 3 and 10 times the standard deviation of noise to the lower point of standard level [45] for both compounds (Appendix A). Two quality control samples (QC) at two different concentration levels were used for assessing the within-day (Appendix A) and between-day variation (Appendix A). The within-day imprecision (Coefficient of Variability (CV) %) and inaccuracy (%) were calculated analysing, in the same analytical run, each level of QC samples 6 times (Appendix A). The between-day imprecision (CV %) and inaccuracy (%) were calculated analysing each level of QC samples once a day, for 5 days (Appendix A).

### 2.6. Data Analysis

Instrument control, data acquisition of MS spectra acquired by UHPLC-Orbitrap, were performed using the associated Excalibur software version 4.0. Instrument control, data acquisition, and processing of MS spectra acquired by UPLC-QTrap were performed using the associated Analyst and MultiQuant Software version 1.6 and 3.0.2, respectively. Raw UPLC-Orbitrap data were converted to mzML file format using the open-source software ProteoWizard [46]. Data processing was carried out using the software MZMine 2 [47]. The data processing workflow included: (1) mass detection; (2) ADAP chromatogram builder [48]; (3) smoothing; (4) deconvolution; (5) deisotoping; (6) alignment (RANSAC Aligner); (7) gap filling. More details about the parameters are available in Appendix A. Statistical analysis that included Principal Component Analysis (PCA), Partial Least Squares-Discriminant Analysis (PLS-DA) and Hierarchical Clustering were conducted using MetaboAnalyst© 5.0 [49,50]. A normalization to sample median, square root data transformation and autoscaling was applied. Metlin [51], NpAtlas [52] and *m*/*z* mine [53] databases were used for dereplication attempt.

## 3. Results

### 3.1. Extraction of Phaeoacremonium Culture Filtrates

Four isolates of *Phaeoacremonium italicum*, three isolates of *P. alvesii* and two isolates of *P. rubrigenum* were tested for their ability to produce phytotoxins in vitro, as reported in the material and methods section. The results for all these samples are reported in Table 2. The highest yield in organic extract was obtained in the range between 7.13 and 28.15 mg per 40 mL filtrate when the culture filtrates were extracted at acid pH (Table 2).

### 3.2. Phytotoxic Bioassays

Symptoms on cotyledons treated with culture filtrates appeared 5 days after the inoculation. Necrotic spots, browning of cotyledon veins, and irregular discoloured areas were observed (Figure 1). In particular, among the *P. italicum*, the isolate Pm59 showed both necrotic spots and browning of veins symptoms, while isolates CBS 137763, Pm50M and Pm45 showed just browning of veins or necrotic spots. Whereas, *P. alvesii* isolates CBS 113590, CBS 408.78 and CBS 729.97, showed necrotic spots and browning of veins. *P. rubrigenum* isolates CBS 498.94 and CBS 112046 showed light necrotic spots and irregular discoloured areas. No symptoms were observed on cotyledons assayed with Czapek broth and sterile distilled water (Figure 1).

The symptoms that occurred on cotyledons assayed with organic extracts at 3 mg/mL started to appear at the fifth day after inoculation, and the toxicity severity increased during the following 3 days. All the *Phaeoacremonium* species extracts caused necrotic spots on cotyledons regardless of the concentration and the pH of extraction, while the appearance of necrotic areas, discoloured areas and chlorotic ring depended on species, pH, and concentration (Table 3). The symptoms caused by the extracts of *P. italicum* Pm50M and Pm45, *P. alvesii* CBS 729.97 and *P. rubrigenum* CBS 498.94 varied depending on extraction pH values (UpH and pH 2). In detail, the extracts at unmodified pH (UpH) of *P. italicum* Pm50M and *P. alvesii* CBS 729.97 caused necrotic spots surrounding the puncture, irregular discoloured areas and marginal necrotic areas, whereas only necrotic spots were observed on cotyledons treated with pH 2 extracts. The extracts at pH 2 of *P. italicum* isolate Pm45 and *P. rubrigenum* isolate CBS 498.94 showed necrotic spots surrounding the puncture, irregular discoloured areas and marginal necrosis, while necrotic spots and irregular discoloured areas were observed on cotyledons treated with UpH extracts. The remaining 10 extracts gave the same symptoms regardless of the pH of extraction (Table 3; Figure 1). Similar symptoms were observed for the culture filtrates treated at 1.5 mg/mL, even though they were less pronounced. No symptoms were observed on cotyledons assayed with Czapek broth and MeOH (4%).

The toxicity severity (*TS*) scores, calculated as reported in the materials and methods section (Equation (1)), varied among species and within isolates. Among *P. italicum* isolates, Pm45 had the highest *TS* score (1.63, UpH), while Pm50M had the lowest *TS* score (1.00, pH 2). Among the *P. alvesii* isolates, CBS 113590 had the highest *TS* score (2.13, UpH), while CBS 729.97 had the lowest *TS* value (1.00, pH = 2). Among the *P. rubrigenum* isolates, CBS 498.94 had the highest *TS* score (2.00, pH 2), while by CBS 112046 had the lowest *TS* value (1.25, pH 2).

### 3.3. Dereplication of Scytalone *(**1**)* and Isosclerone *(**2**)*

For the dereplication procedure, standard solutions of scytalone (**1**) and isosclerone (**2**) were analysed by UHPLC-Orbitrap in polarity switching mode. The best ionization condition for both compounds was found in the negative mode, thus the HRMS data, the MS/MS fragments, and retention time (RT), were annotated (Table 1). After filtration, 20 extracts, including 18 extracts of the fungal isolates and 2 extracts of the culture medium, were analysed in duplicate under the same conditions of the standards. Figure 2 shows examples of the Total Ions Chromatograms (TICs) of the extracts at unmodified pHs and under acid conditions (pH 2).

The raw spectra were then visualised by Excalibur software that allowed rapid dereplication of scytalone (**1**), *m*/*z* 193.0491 (4.60 min) [M − H]^−^, and isosclerone (**2**), *m*/*z* 177.0541 (4.69 min) [M − H]^−^, by matching retention times, HRMS, and MS/MS data manually. The presence of compound **1** was confirmed in the extracts of both *P. alvesii* isolates CBS 729.97 and CBS 408.78, *P. italicum* isolates CBS 137763, Pm45, Pm59 and *P. rubrigenum* isolate CBS 112046. Furthermore, compound **1** was mainly detected in the UpH extracts (Table 4). Isosclerone (**2**) was detected in the extracts of *P. italicum* isolates Pm45 and CBS 137763, *P. alvesii* isolate CBS 408.78 and *P. rubrigenum* isolate CBS 498.94. Compound **2** was detected only in the in the UpH extracts.

### 3.4. Quantification of Scytalone *(**1**)* and Isosclerone *(**2**)*

The scheduled MRM method (sMRM) consists in a short analytical cycle of 10 min (Appendix A). Two different calibration curves were developed for scytalone (**1**) and isosclerone (**2**). The LODs and the r values are reported in Appendix A. Before proceeding with the analysis of the extracts, two levels of QC samples were prepared to evaluate imprecision and inaccuracy of the method. The within-day imprecision (CV%) and inaccuracy (%) were calculated by analysing, in the same analytical run, each level of QC samples six times. The imprecision varied from 2.2% to 4.2% for scytalone (**1**) and 1.9% to 3.1% for isosclerone (**2**). The inaccuracy (%) ranged from −5.7% to 5.5% for scytalone (**1**), and from −0.8% to −0.8% for isosclerone (**2**) (Appendix A). The between-day imprecision (CV%) and inaccuracy (%) were calculated analysing each level of QC samples once a day, for 5 days. The imprecision is estimated from (CV%) 5.6% to 7.6% for scytalone (**1**), from 2% to 6.3% for isosclerone (**2**); the inaccuracy was estimated from (%) 1.4% to 1.75% for scytalone (**1**) and from 1.8% to 2.3% for isosclerone (**2**) (Appendix A). After the validation the 20 extracts were analysed in triplicates and the quantitative data analysed by Multiquant software. The concentration of scytalone (**1**) was higher in all the UpH extracts (Table 3) and was >LOQ in the following isolates: *P. italicum* Pm45 and CBS 137763, *P. alvesii* isolates CBS 729.97 and CBS 408.78 and *P. rubrigenum* isolates CBS 112046. *P. italicum* Pm45 UpH extract showed the higher concentration of **1** (11.90 μg/mL ± 1.05), while the lowest concentration (1.00 μg/mL ± 0.12) was found in *P. rubrigenum* isolate CBS 112046 (Table 4). Isosclerone (**2**) was quantifiable only in *P. italicum* isolates Pm45 and CBS 137763. The highest concentration (86.67μg/mL ± 5.08) of **2** was found in *P. italicum* CBS 137763. Finally, only *P. italicum* isolates CBS 137763 and Pm45 produced both compounds in quantifiable amounts.

### 3.5. Statistical Analysis Using MetaboAnalyst

MzMine 2 was used for the processing of UHPLC-Orbitrap spectra as described in the materials and method section. All the parameters used for the processing are reported in Appendix A. The obtained peak lists were analysed by MetaboAnalyst 5.0 using the statistical analysis module.

PCA was used to observe an overview of variance between the extracts at different pH and secondary metabolite compositions. The PCA score plot showed the *Phaeoacremonium* species crude extracts clustered according to the extraction condition (Figure 3a). However, the PCA showed no differences among or between the species. This result was also confirmed by the hierarchical clustering, shown as a dendrogram in Appendix A. Thus, even though it was possible to cluster the organic extracts depending on the extracting conditions, it was not possible to separate the species according to their metabolic profile. The box plots of scytalone (**1**), *m*/*z* 193.0491 (4.60 min) [M − H]^−^, and isosclerone (**2**), *m*/*z* 177.0541 (4.69 min) [M − H]^−^, reported in Figure 3b, show a high concentration of these two compounds in the UpH extracts. These data further confirm the results arose from the quantitative analysis (Table 4).

Partial Least Squares-Discriminant Analysis (PLS-DA) was carried out to investigate the significant differences among metabolites extracted at different pH, and to further explore possible differences among the species. The PLS-DA score plot (Figure 3c) confirms the PCA results: the crude extracts were again grouped according to the extraction condition. The statistical model showed a good predictivity having Q2 and R2 values, obtained using Pareto scaling at five components of 1 and 0.9, respectively. The variable importance of projection (VIP) score scatter plot indicating the top 15 features, responsible for the clustering of the groups, was shown in Figure 3d. A total of 13 out of 15 features of the top VIP score were most abundant in the UpH extracts. Discriminating features, VIP scores >1.55, for crude extracts included *m*/*z* 389.2514 (6.53 min, [M − H]^−^, C_16_H_33_N_6_O_5_), *m*/*z* 303.1796 (5.31 min, [M − H]^−^, C_13_H_25_N_3_O_5_), *m*/*z* 387.1999 (6.41 min, [M + H]^+^, C_17_H_29_N_3_O_7_) and 249.0754 (4.75 min, [M + H]^+^, C_13_H_12_O_5_). Putative identification was attempted using different online databases: Metlin, NpAtlas and *m*/*z* cloud, however no proper matches were found.

### 3.6. Selection of Phaeoacremonium Isolates for Further Investigation

Several unknown compounds can be spotted in the untargeted analysis but the comparison of their MS and MS^2^ data with those reported on web-based databases was unsuccessful. In order to start a first characterization of these compounds, one isolate of each species was selected for additional in vitro growth. All the fungal organic extracts showed phytotoxicity in the assay condition, thus the selection of the fungal isolates was carried out, also comparing the results of multivariate statistical analysis and the metabolic profile obtained by UHPLC-Orbitrap.

Thirteen out of fifteen of the top features in the VIP score (Figure 3d) were most abundant in the UpH extracts. Therefore, the metabolic profiles of organic extracts of fungal isolates at UpH, grouped by species (Figure 4a–c), were compared to select the most promising samples. According to the biological assay, isolate Pm45 had the highest TS score (1.63) among the *P. italicum*. As shown in Figure 4a, this isolate had also the richest metabolic profile among this species. Interestingly the peak at 3.54 min showed the same high-resolution mass of scytalone (**1**) but different MS^2^ pattern, which is suggestive of a possible structural isomerism. The structure elucidation of this compound is currently under investigation. Within the *P. alvesii* species, the isolate CBS 113,590 had the highest TS score (2.13). The isolate CBS 729.97 showed the second TS score (1.50) and produced the highest amount of scytalone (**1**). This isolate was also characterized by the richest metabolic profiles, with a very diagnostic peak at 10.76 and *m*/*z* 167.0127 that is currently under study for structure elucidation (Figure 4b). Within the *P. rubrigenum* species the isolate CBS 498.94 had the highest TS score (2.00) and the richest metabolic profile. In this sample, the most abundant peak occurred at 5.89 min and showed a molecular mass at *m*/*z* 198.0173. As for the other unknown compounds reported above, this product is also currently under investigation (Figure 4c). Figure 4d shows a comparison of the chromatograms of the selected isolates, some of the peaks were shared between the isolates, even though with different abundance. Other peaks seem to be specific for a selected isolate (Figure 4d).

## 4. Discussion

To the best of our knowledge, this is the first study reporting the dereplication and quantification of scytalone (**1**) and isosclerone (**2**) produced in vitro by *Phaeoacremonium* species applying untargeted and targeted LC–MS based metabolomics approaches. Prior to the present analysis, no data were available in literature on the secondary metabolites produced by *P. italicum*, *P. alvesii* and *P. rubrigenum*. Moreover, the untargeted metabolic profiles of their organic extracts were investigated for the presence of other secondary metabolites. Two different pH conditions, unmodified pH and pH 2 were used for the extraction of secondary metabolites. The number of organic extracts obtained results higher for pH 2 according to previous reported data in literature for other pathogenic fungi involved in grapevine trunk diseases [31,32]. The pH is a parameter that can affect the extraction of secondary metabolites, due to their diverse nature and physicochemical properties. Consequently, an extraction condition suitable for one chemical class may be unsuitable for another.

The choice of the best extraction condition plays a dominant role in the comprehensiveness and the representativeness of the metabolite profile obtained, and sometimes could be challenging [54]. This aspect is fundamental especially when multi-omics studies want to be applied. Different extraction procedures optimized for different biomolecules are needed, and a flexible extraction method providing robust and reliable recovery of the molecular components is advisable [55].

Both culture filtrates and organic extracts used in bioassays produced different phytotoxicity degrees, although the symptom kinds were similar. In particular, necrotic spots, browning of cotyledon veins, and irregular discoloured areas were more intense in the culture filtrates rather than in the organic extracts. The higher phytotoxicity of the culture filtrates could be related to the presence of high molecular weight phytotoxins in the culture filtrates, which could have a synergist effect intensifying the observed symptoms [56], and/or to the presence of active fungal propagules passed through out the filter paper. Indeed, toxic exopolysaccharides and polypeptides have been isolated from *Phaeoacremonium minimum,* one of the main causal agents of Esca complex culture filtrate [57,58]. Thus, future studies could be conducted to investigate high molecular weight phytotoxins produced by *Phaeoacremonium* species to understand their role in pathogenicity.

*Phaeoacremonium minimium* is also known to produce two phytotoxic naphthalenone pentaketides: scytalone (**1**) and isosclerone (**2**) [29]. They were also identified in liquid culture filtrates of *Phaeomoniella chlamydospora* also involved in Esca complex [30]. For this reason, **1** and **2** were dereplicated in the organic extracts of the three selected *Phaeoacremonium* species by matching retention times, and HRMS and MS/MS data, with pure standard samples.

Dereplication strategies are needed to screen the crude extracts for the presence of known compounds before isolation efforts are initiated. In fact, re-isolation of known compounds is a crucial problem, wasting time and resources, in the discovery of new biologically active compounds from natural sources [42]. To tackle this problem, modern dereplication strategies have been developed. They use high throughput screening techniques, such as tandem mass spectrometry, coupled with bioinformatics tools and databases, both in house built and web based [59,60]. The progress of metabolomics and molecular networking also helped in the development dereplication strategies [61].

Scytalone (**1**) was dereplicated in *P. italicum* isolates Pm45, Pm59 and CBS 137763, *P. alvesii* isolates CBS 729.97 and CBS 408.78, and *P. rubrigenum* isolate CBS 112046. Isosclerone (**2**) was dereplicated in *P. italicum* isolate Pm45 and CBS 137763, *P. alvesii* isolate CBS 408.78 and *P. rubrigenum* isolate CBS 498.94.

To gain further insights into the production of **1** and **2** a quantitative targeted LC–MS method has been developed on UPLC-QTrap and then validated. QTrap together with triple quadrupole (QqQ) are the work horses in targeted approaches. Both instruments rely on MRM (Multi Reaction Monitoring) as their most sensitive and reliable method in the quantitation of metabolites [62]. As previous attempts to quantify scytalone (**1**) and isosclerone (**2**) from culture filtrates of *P. minimum* by LC–MS failed [63], this study reports for the first time a valid targeted MS method for their quantification in in vitro culture of *Phaeoacremonium* species. In details, scytalone (**1**) was detectable in both pH extracts, although the highest amount was found in the unmodified pH, while isosclerone (**2**) was measurable only in the UpH extracts. This difference could be explained considering the chemical features of scytalone (**1**). Compound **1,** having two phenolic groups, is more acid then isosclerone, thus could be extracted also at pH 2, although in smaller quantities than UpH. These results highlighted that the pH is a key parameter for the extraction of **1** and **2**, and, with high probability, also for close related metabolites produced by this genus [28]. *P. italicum* isolate Pm45 produced the highest amount of scytalone (**1**). A quantifiable amount of **1** was also detected in *P. italicum* CBS 137763, *P. alvesii* isolates CBS 729.97 and CBS 408.78, and *P. rubrigenum* isolate CBS 112046. Isosclerone (**2**) was measurable only in *P. italicum* isolates Pm45 and CBS 137763. Furthermore, combining these quantitative data with the biological assay results, no correlation between the amount of **1** and **2** and toxicity severity scores was found, because they were independent from the amount of scytalone and isosclerone found in the extracts. Hence, also in this case, synergistic or antagonistic effects [56] could be present in the extracts and should be studied.

These results point out the importance of structural identification of the components of the organic extracts. The pre-treatment of the untargeted LC–MS data was carried out by MzMine 2.0 [47], while the statistical analysis was carried out by MetaboAnalyst [50]. The principal component analysis (PCA) clustered the crude extracts into two groups according to the extraction pH values. These data further confirm the importance of pH value as a key parameter for the extraction process. On the contrary, no differences were highlighted in the extracts of distinct species. Moreover, the PCA confirmed the highest amount of scytalone (**1**) and isosclerone (**2**) in the UpH. Partial Least Squares-Discriminant Analysis (PLS-DA) further confirmed the previous results. Organic extracts clustered according to the pH values, but regardless of the species. Although the multivariate analysis did not show differences between the selected species, this workflow could be easily applied to compare the metabolic profile of other pathogenic fungi involved in the Esca complex or, in general, other fungi associated with GTDs. The main goal of these studies could be the identification of potential species-specific biomarkers.

The differences between the unmodified UpHs and pH 2 were mainly due to unknown metabolites; the attempt to dereplicate features having a VIP score >1.55 using online database failed. The annotation of small molecules remains a major pitfall in untargeted LC–MS-based metabolomics. The association of a detected individual signal to its corresponding metabolite identities is often performed by querying detected signals against one or several experimental web-based databases. Outstanding progress has been made in the last decade in growing the number of metabolites in databases; however, these databases are far from being comprehensive. Researchers involved in in spectroscopic characterization play an important role in filling this gap, and they should be prone to uploading the collected spectra in metabolomics databases [64].

Only by elucidating unknown metabolites is it possible to biologically interpret complex systems, gaining insight into the chemical-ecology of pathogenic fungi. Assignment of the chemical structure to fungal unknown secondary metabolites could be beneficial for investigations that deal with fungal pathogenic behavior [65], host-pathogen interaction [66,67,68] or, more in general, that use multi-omics approaches [69]. Consequently, three isolates (Pm45; CBS 729.97; CBS 498.94), one for each of *Phaeoacremonium* species, were selected according to the toxicity severity score, multivariate statistical analysis and untargeted metabolic profile, to be grown in vitro in larger amounts (4 to 8 L), to achieve the isolation and chemical (by spectroscopic method) and biological characterization of the unknown secondary metabolites. The spectroscopic data collected will be uploaded on the most used metabolomic database.

## 5. Conclusions

Investigations on the bioactive natural products produced by fungi have a significant impact in many scientific research fields, from plant pathology to organic chemistry. Using novel tools such as metabolomics can help to better understand and interpret complex biological phenomena, such as host–pathogen interaction. Scytalone (**1**) and isosclerone (**2**), and a few other naphthalenone pentaketides have been isolated from *Phaeoacremonium* species to date [28]. However, the untargeted metabolic profiling disclosed several unknown compounds that could be related to other secondary metabolites that could play a role in the life cycle of this species. Moreover, species of *Phaeoacremonium* have been reported to be responsible for infections in humans [11], thus investigating the role of secondary metabolites in pathogenesis could also be noteworthy in medicine applications. Indeed, as described by the One Health concept [70], human health is linked to the health of the ecosystems of which we are a part.

## Figures and Tables

**Figure 1 jof-08-00055-f001:**
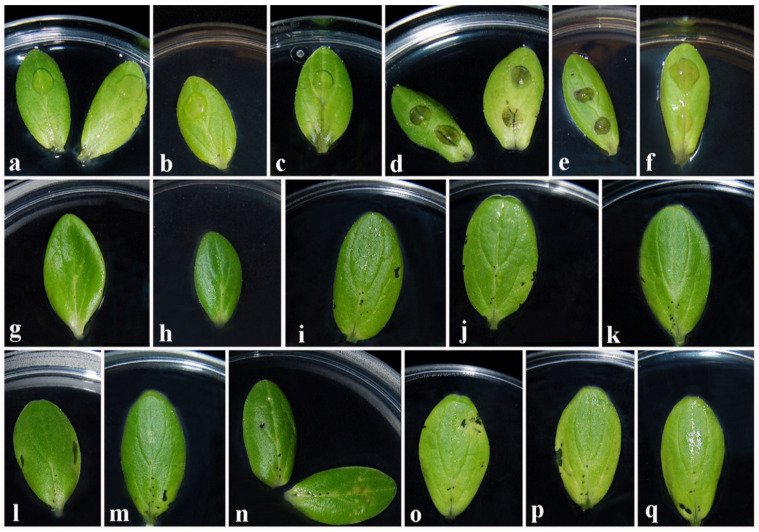
Bioassays carried out with culture filtrate (**a**–**f**) and organic extracts (**g**–**q**) of different isolates of three *Phaeoacremonium* species. (**a**) sterile distilled water; (**b**) Czapek broth; (**c**) browning of cotyledon veins caused by culture filtrate of CBS 137763 and Pm50M (*P. italicum*); (**d**) necrotic spots and browning of veins caused by culture filtrate of Pm59 (*P. italicum*) and all the three isolates of *P. alvesii*; (**e**) necrotic spots caused by culture filtrate of Pm45 (*P. italicum*); (**f**) light necrotic spots and irregular discoloured areas caused by culture filtrate of both isolates of *P. rubrigenum*; (**g**) methanol 4%; (**h**) Czapek broth; (**i**) symptoms caused by organic extracts of CBS 137763 (*P. italicum*); (**j**) symptoms caused by organic extracts of Pm50M (*P. italicum*); (**k**) symptoms caused by organic extracts of Pm45 (*P. italicum*); (**l**) symptoms caused by organic extracts of Pm59 (*P. italicum*); (**m**) symptoms caused by organic extracts of CBS 729.97 (*P. alvesii*); (***n***) symptoms caused by organic extracts of CBS 113590 (*P. alvesii*); (**o**) symptoms caused by organic extracts of CBS 408.78 (*P. alvesii*); (**p**) symptoms caused by organic extracts of CBS 498.94 (*P. rubrigenum*); (**q**) symptoms caused by organic extracts of CBS 112046 (*P. rubrigenum*).

**Figure 2 jof-08-00055-f002:**
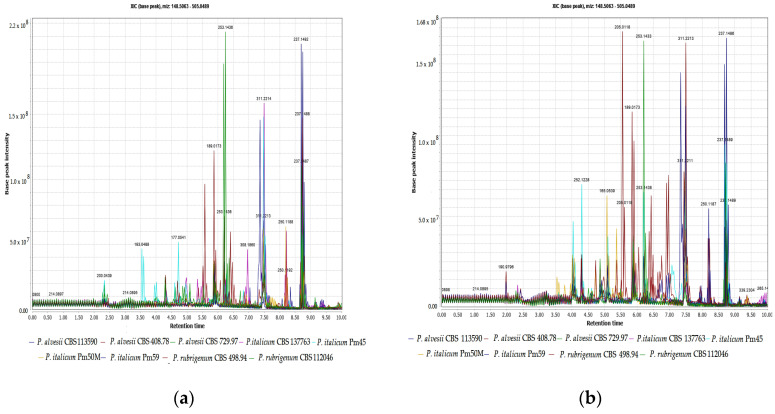
(**a**) Total Ions Chromatograms (TICs) of organic extracts of *Phaeoacremonium* spp. and culture media at UpHs; (**b**) (TICs) of organic extracts of *Phaeoacremonium* spp. and culture media at pH 2.

**Figure 3 jof-08-00055-f003:**
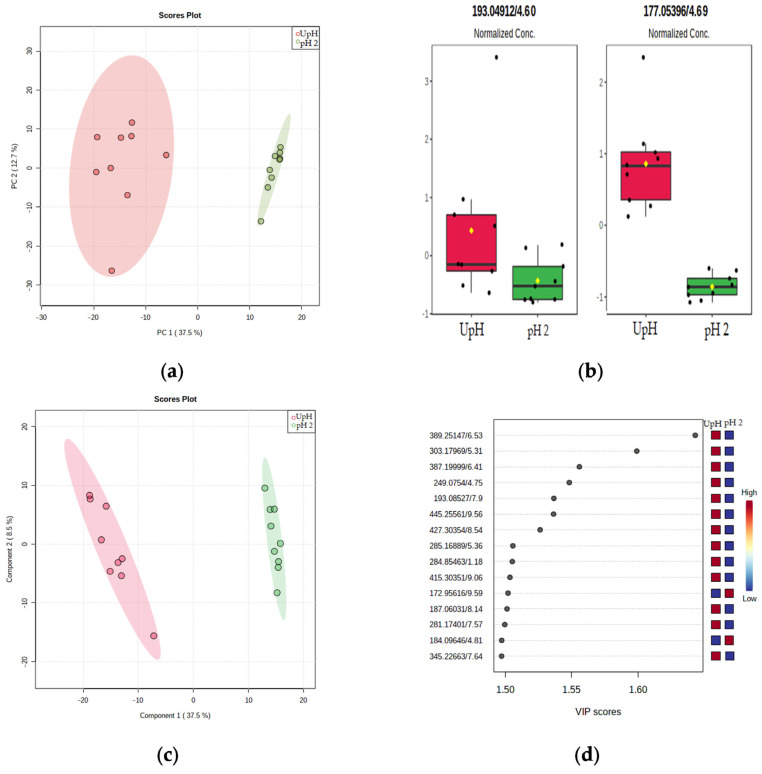
(**a**) PCA scores plot between PC1 and PC2. The explained variances are shown in brackets; (**b**) box plots reporting the normalized concentration for scytalone (**1**), *m*/*z* 193.0491 (4.60 min), and isosclerone (**2**), *m*/*z* 177.0541 (4.69 min); (**c**) PLS-DA scores plot between PC1 and PC2. The explained variances are shown in brackets. (**d**) Important features identified by PLS-DA. The colored boxes on the right indicate the relative concentrations of the corresponding metabolite in unmodified pH group (UpH) and acid pH group (pH 2).

**Figure 4 jof-08-00055-f004:**
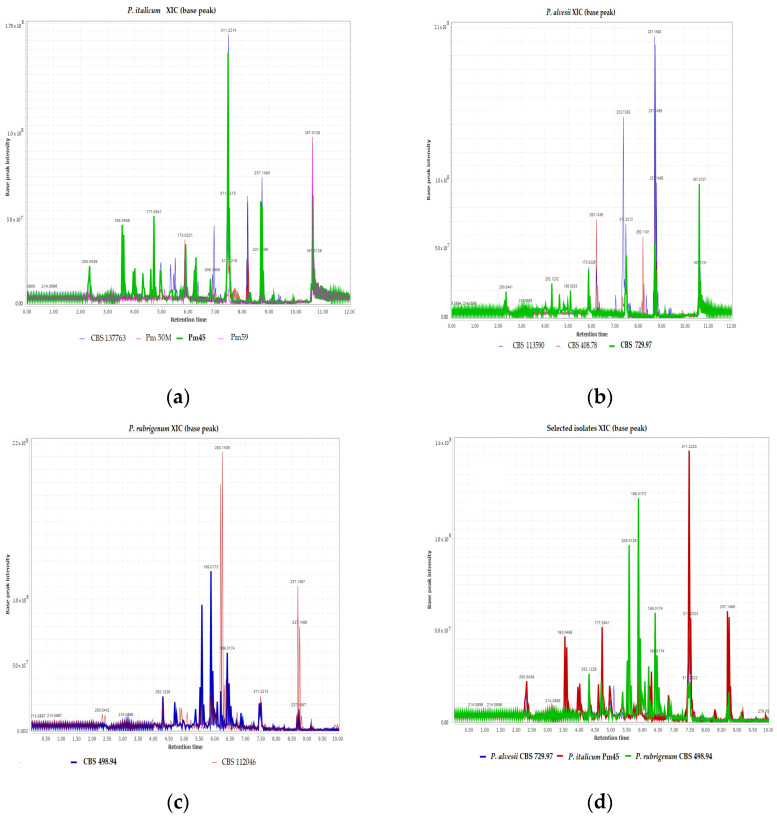
(**a**) *Phaeoacremonium italicum* isolates TICs comparison; (**b**) *P. alvesii* isolates TICs comparison; (**c**) *P. rubrigenum* isolates TICs comparison; (**d**) *Phaeoacremonium* species and isolates selected for further investigation.

**Table 1 jof-08-00055-t001:** Optimized Q1 mass, transitions, and parameters for LC-MS/MS analysis.

Compounds	Precursor Ion (*m*/*z*)	Product Ion (*m*/*z*)	RT 1 ^a^ (min)	Qtrap Parameter			RT 2 (min)
				DP ^b^	CE ^c^	CXP ^d^	
scytalone (**1**) 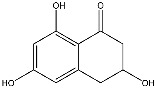	193.0491 [M − H]^−^	151.0382 (Quantifier) 123.0426 (Qualifier)	4.60	−70 −70	−30 −32	−15 −13	4.35
isosclerone (**2**) 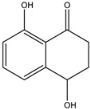	177.0541 [M − H]^−^	159.0427 (Quantifier) 149.0582 (Qualifier)	4.68	−75 −75	−23 −21	−14 −11	5.28
2-hydroxy-1,4-naphtoquinone (**3**) 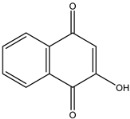	173.0234 [M − H]^−^	145.0272	5.85	−55	−23	−11	5.58

^a^ RT = retention time ^b^ DP = declustering potential; ^c^ CE = collision energy; ^d^ CXP = Collision cell exit potential.

**Table 2 jof-08-00055-t002:** Selected *Phaeoacremonium* species and isolates. Amount of fungal filtrates organic extract (mg) at different pH.

Fungal Species	ID Isolate	pH Culture Filtrates	Extract UpH ^a^ (mg)	Extract pH 2 (mg)
*P. italicum*	CBS 137763	8.5	4.88	7.13
	Pm50M	8.4	8.90	8.01
	Pm45	8.2	4.18	8.26
	Pm59	6.1	15.99	28.15
*P. alvesii*	CBS 113590	8.8	7.11	8.55
	CBS 729.97	6.3	7.23	16.57
	CBS 408.78	8.8	4.41	6.16
*P. rubrigenum*	CBS 112046	4.4	4.8	8.70
	CBS 498.94	7.9	10.92	9.56
Culture medium (Czapek)		6.2	2.60	3.50

^a^ Unmodified pH.

**Table 3 jof-08-00055-t003:** Phytotoxic bioassay carried out on *Cucumis sativus* cotyledons with organic extracts from *Phaoacremonium* isolates.

			Symptoms Observed on Cotyledons after 8 Days									
			Necrotic Areas		Discoloured Areas		Chlorotic Ring		Necrotic Spots		Toxicity Severity (*TS*)	
	Organic Extract Concentrations		3 mg/mL	1.5 mg/mL	3 mg/mL	1.5 mg/mL	3 mg/mL	1.5 mg/mL	3 mg/mL	1.5 mg/mL	3 mg/mL	1.5 mg/mL
Fungal Species	ID Isolates	pH Values										
*P. italicum*	CBS 137763	UpH ^a^	+	+	−	−	−	−	+	+	1.25	1.13
		pH 2	+	+	−	−	−	−	+	+	1.25	1.25
	Pm50M	UpH	+	+	+	−	−	−	+	+	1.25	1.25
		pH 2	−	−	−	−	−	−	+	+	1.00	1.25
	Pm59	UpH	+	+	+	−	−	−	+	+	1.50	1.38
		pH 2	+	−	+	−	−	−	+	+	1.13	1.13
	Pm45	UpH	−	−	+	+	−	−	+	+	1.63	1.13
		pH 2	+	+	+	−	−	−	+	+	1.13	1.00
*P. alvesii*	CBS 408.78	UpH	+	−	+	−	+	−	+	+	1.38	1.00
		pH 2	+	+	+	+	+	−	+	+	1.38	1.13
	CBS 729.97	UpH	+	+	+	−	+	−	+	+	1.38	1.25
		pH 2	−	−	−	−	−	−	+	+	1.50	1.00
	CBS 113590	UpH	+	+	+	+	−	−	+	+	2.13	1.25
		pH 2	+	+	+	+	+	+	+	+	1.63	1.25
*P. rubrigenum*	CBS 112046	UpH	+	+	+	−	−	−	+	+	1.38	1.25
		pH 2	+	+	+	−	−	−	+	+	1.25	1.13
	CBS 498.94	UpH	−	−	+	−	−	−	+	+	1.38	1.13
		pH 2	+	+	+	+	−	−	+	+	2.00	1.13
Control	Czapek	UpH	−	−	−	−	−	−	−	−	0.00	0.00
		pH 2	−	−	−	−	−	−	−	−	0.00	0.00

^a^ = unmodified pH; +/− symptom presence/absence; *TS*, mean value of toxicity empiric scale.

**Table 4 jof-08-00055-t004:** Identification and quantification of scytalone (**1**) and isosclerone (**2**) by LC-MS/MS analysis.

Fungal Species	Isolate	pH	Orbitrap		Qtrap	
			Scytalone	Isosclerone	Scytalone (μg/mL ± SD)	Isosclerone (μg/mL ± SD)
*Phaeoacremonium italicum*	Pm50M	UpH ^a^	-	-	<LOD ^c^	<LOD
		pH 2 ^b^	-	-	<LOD	<LOD
	Pm45	UpH	+	+	11.90 ± 1.05	65.82 ± 3.86
		pH 2	-		<LOD	<LOD
	Pm59	UpH	+	-	<LOD	<LOD
		pH 2	-	-	<LOD	<LOD
	CBS 137763	UpH	+	+	5.67 ± 0.50	86.67 ± 5.8
		pH 2	+	+	3.64 ± 0.28	<LOQ ^d^
*P. alvesii*	CBS 729.97	UpH	+	-	10.43 ± 0.98	<LOQ
		pH 2	+	-	4.07 ± 0.32	<LOD
	CBS 408.78	UpH	+	+	8.52 ± 1.02	<LOQ
		pH 2	+	-	3.62 ± 0.12	<LOD
	CBS 113590	UpH	-	-	<LOD	<LOD
		pH 2	-	-	<LOD	<LOD
*P. rubrigenum*	CBS 112046	UpH	+	-	1.00	<LOD
		pH 2	-	-	<LOD	<LOD
	CBS 498.94	UpH	-	+	<LOD	<LOQ
		pH 2	-	-	<LOD	<LOD
Culture medium (Czapek)		UpH	-	-	-	-
		pH 2	-	-	-	-

^a^ UpH = extracted at same pH of the culture filtrate; ^b^ pH 2 = extracted after acidification; ^c^ <LOD = the amount is below the limit of detection ^d^ <LOQ = the amount is below the limit of quantification.

## Data Availability

The data presented in this study are contained within the article and Appendix A.

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
