# Peer review of "Untargeted and Targeted LC-MS/MS Based Metabolomics Study on In Vitro Culture of Phaeoacremonium Species"

_jof, 2022, doi:10.3390/jof8010055_

Round 1

Reviewer 1 Report

No changes needed.

Author Response

We thank the Reviewer for the very positive comments 

Reviewer 2 Report

The authors in their article Untargeted and targeted LC-MS / MS based metabolomics study on in vitro culture of Phaeoacremonium species have successfully demonstrated the use of modern analytical methods (LC-MS / MS) in combination with classical analytical methods to successfully characterize different Phaeoacremonium species.

The authors have developed a very successful experimental design and the profiling of high quality chromatograms with mass spectrometry and chemical compounds has successfully shown which compounds are directly related to their activity.

The work needs to be supplemented with statistical data processing, as no standard deviations (SD) are given in the tables and figures to assess the significance of the results.

Reviewer 3 Report

The manuscript entitled “Untargeted and targeted LC-MS/MS based metabolomics study on in vitro culture of Phaeoacremonium species” by Pierluigi Reveglia et al. reports a LC-MS/MS base metabolomics approach to identify secondary metabolites produced by pathogenic Phaeoacremonium spp. Manuscript needs major English edition and presentation of the finding more precisely. Comments/ suggestions attached.  

Round 2

Reviewer 3 Report

Authors revised the manuscript according to the Reviewers' comments and suggestions. But still major English correction is needed. It is not the Reviewers' job to correct the English Language. But still I have highlighted in the revised manuscript. I am not gonna give separate explanations to the highlighted points. Its Authors duty to correct it. But I'll highlight the other corrections as follows. 

Abstract: use of 2 different MS methods not clear. make a statement!

Text:

88 - remove worldwide

97 - fungi 

119 - medicine and agriculture

130 - and

303-305 - space between unit and the number (3 g/L)

326 - the 

equation 1 - insert a clear one

364 - UPLC solvent gradient is wrong! 10% to 80% ACN in 0.1% formic acid!

367 - either mL or ml? use micro symbol 

373 - mass range

374 - 1 x 10^6

537-547 - one —> extract, many —> extracts!!

Page 11 - check font size

Page 13 - Untargeted analysis section. There are 3 unidentified peaks reported. The data should present more attractively, since it looks like the is incomplete or still on going. Compounds are still not identified and data is missing. Whether M/Z values reported are [M+H]+ or [M-H]-, are there any Na adducts, predicted molecular formulae, etc can add to this section. What are the online databases Authors used to dereplicate the compounds? Try NpAtlas if there is any luck with those unidentified peaks. 

Still I am not convinced about the "Untargeted Metabolomics" section of the manuscript. Though the Authors commented that they followed the reference article of Goodcare et al., the manuscript is lack of untargeted metabolites produced by those fungi. The Metabolomics data presented is not in the standard level of reporting. Please report the Metabolomics dada according to the Metabolomics Standard Initiative levels (DOI 10.1007/s11306-007-0082–2) 

Author Response

Authors revised the manuscript according to the Reviewers' comments and suggestions. But still major English correction is needed. It is not the Reviewers' job to correct the English Language. But still I have highlighted in the revised manuscript. I am not gonna give separate explanations to the highlighted points. Its Authors duty to correct it. But I'll highlight the other corrections as follows. 

Abstract: use of 2 different MS methods not clear. make a statement!

Answer: The following statement has been added: “High resolution mass spectrometer UHPLC-Orbitrap was used for the untargeted profiling and dereplication of secondary metabolites. A sensitive multi reaction monitoring (MRM) method for the absolute quantification of scytalone and isosclerone was developed on a UPLC-QTrap”

Text:

88 - remove worldwide

Answer: worldwide has been removed

97 - fungi 

Answer: “fungus” has been modified with fungi

119 - medicine and agriculture

Answer: “medicinal and agricultural” have been modified with “medicine and agriculture”

130 – and

Answer: accordingly modified

303-305 - space between unit and the number (3 g/L)

Answer: the space has been added

326 - the equation 1 - insert a clear one

Answer: the equation 1 has been modified and replaced with a clear one

364 - UPLC solvent gradient is wrong! 10% to 80% ACN in 0.1% formic acid!

Answer: the gradient has been corrected

367 - either mL or ml? use micro symbol 

Answer: We prefer to report the capital letter L, as also suggested by the International System of Unit to avoid possible confusion between the numeral 1 (one) and the lower-case letter l (el). (https://www.bipm.org/documents/20126/41483022/si_brochure_8.pdf)

373 - mass range

Answer: mass has been added

374 - 1 x 10^6

Answer: accordingly modified

537-547 - one —> extract, many —> extracts!!

Answer: We thank the reviewer for this comment, we checked and corrected, where needed, the word extract.

Page 11 - check font size

Answer: We checked, and corrected the font size

Page 13 - Untargeted analysis section. There are 3 unidentified peaks reported. The data should present more attractively, since it looks like the is incomplete or still on going. Compounds are still not identified and data is missing. Whether M/Z values reported are [M+H]+ or [M-H]-, are there any Na adducts, predicted molecular formulae, etc can add to this section. What are the online databases Authors used to dereplicate the compounds? Try NpAtlas if there is any luck with those unidentified peaks. 

Answer. The required information about the retention time, M/Z value, ion type ([M+H]+ or [M-H]-) data have been added and discussed as required (this data are sufficient to describe unknown (level 4) metabolites as reported in the Metabolomics Standard Initiative manuscript suggested by the reviewer). Furthermore, information about the databases used to try their dereplication (Metlin, NpAtals and Mz/Cloud) have been added. However, it was not possible to identify the compounds, because no proper match were found in the databases.

Still I am not convinced about the "Untargeted Metabolomics" section of the manuscript. Though the Authors commented that they followed the reference article of Goodcare et al., the manuscript is lack of untargeted metabolites produced by those fungi. The Metabolomics data presented is not in the standard level of reporting. Please report the Metabolomics dada according to the Metabolomics Standard Initiative levels (DOI 10.1007/s11306-007-0082–2) 

 Answer: We thank the reviewer for the suggestion manuscript by the Metabolomics Standard Initiative Group, we found it very useful.

However, we do not agree with the sentence “the manuscript is lack of untargeted metabolites produced by those fungi”. Isosclerone and scytalone were identified during the untargeted screening using UPLC-Orbitrap. According to the suggested manuscript, in the section 2.9 Proposed minimum metadata relative to metabolite identification we can read “…A minimum of two independent and orthogonal data relative to an authentic compound analyzed under identical experimental conditions are proposed as necessary to validate non-novel metabolite identifications (e.g. retention time/index and mass spectrum, retention time and NMR spectrum, accurate mass and tandem MS, accurate mass and isotope pattern, full 1H and/or 13C NMR, 2-D NMR spectra).” Isosclerone and scytalone were identified by comparison of the retention time, accurate mass and tandem MS with pure standards, thus they could be defined as level 1 metabolites (Identified Metabolites).

We are aware that other secondary metabolites, for which we report the m/z value, remain unknown. However, in the section 2.10 Proposed minimum metadata relative to reporting of unknown metabolites, we can read “Within most metabolomics datasets, there are typically many unknown analytes, i.e. level 3 and 4 compounds. Obviously, those deemed highly important to the study (In our case we reported the metabolites with the VIP score >1.55) should be rigorously identified according to the metabolite identification discussions above. This is not possible in all cases due to time restrictions or the lack of authentic material for unambiguous assignment. However, these unknown metabolites can often still be differentiated based upon unique experimental data, i.e. spectral or chromatographic features, and it is valuable to systematically report such ‘‘unique unknowns’’ in a meaningful manner”. We thank the reviewer for the suggestion to improve the quality of the reported data for the unknown secondary metabolites. Nevertheless, the putative identification of these secondary metabolites was not possible due to the absence of proper match in the databases. We would like to underline that not all the non-novel secondary metabolites are reported in the databases; indeed, they are far to be comprehensive. For instance, scytalone is covered in NpAtlas but it is absent in M/zmine, while isosclerone is absent in both NpAtlas and Metlin. This highlights the importance to upload data in all available databases, also those of already known secondary metabolites. This pitfall has been further discussed in the manuscript. In conclusion, it is possible to find unknown metabolites during a metabolic profiling analysis, especially when is the first attempt to investigate the secondary metabolites produced by fungal species, such as this case (No previous data were available for secondary metabolites produced by Phaeoacremonium italicum, P. alvesii and P. rubrigenum).
